# Integrated Building Modelling Using Geomatics and GPR Techniques for Cultural Heritage Preservation: A Case Study of the Charles V Pavilion in Seville (Spain)

**DOI:** 10.3390/jimaging10060128

**Published:** 2024-05-27

**Authors:** María Zaragoza, Vicente Bayarri, Francisco García

**Affiliations:** 1School of Engineering in Geodesy, Cartography and Surveying, Universitat Politècnica de València, Camino de Vera, s/n, 46022 Valencia, Spain; 2GIM Geomatics, S.L. C/Conde Torreanaz 8, 39300 Torrelavega, Spain; 3Polytechnic School, Universidad Europea del Atlántico, Parque Científico y Tecnológico de Cantabria, C/Isabel Torres 21, 39011 Santander, Spain; 4Department of Cartographic Engineering, Geodesy and Photogrammetry, Universitat Politècnica de València, Camino de Vera, s/n, 46022 Valencia, Spain

**Keywords:** imaging technologies, geomatics, geophysics, cultural heritage preservation, virtual reconstructions, decision-making models, building information modelling

## Abstract

This paper highlights the fundamental role of integrating different geomatics and geophysical imaging technologies in understanding and preserving cultural heritage, with a focus on the Pavilion of Charles V in Seville (Spain). Using a terrestrial laser scanner, global navigation satellite system, and ground-penetrating radar, we constructed a building information modelling (BIM) system to derive comprehensive decision-making models to preserve this historical asset. These models enable the generation of virtual reconstructions, encompassing not only the building but also its subsurface, distributable as augmented reality or virtual reality online. By leveraging these technologies, the research investigates complex details of the pavilion, capturing its current structure and revealing insights into past soil compositions and potential subsurface structures. This detailed analysis empowers stakeholders to make informed decisions about conservation and management. Furthermore, transparent data sharing fosters collaboration, advancing collective understanding and practices in heritage preservation.

## 1. Introduction

With over a thousand years of history, the Alcazar of Seville (Spain) has emerged as one of the most complex and rich buildings in the world. This architectural marvel offers a captivating journey through the annals of Seville’s past, immersing visitors in the rich tapestry of its history. Recognised as a UNESCO World Heritage Site since 1987 alongside the “Archivo de Indias” and the “Catedral de Santa María de la Sede” [1], the Alcazar embodies the diverse cultural and historical epochs that have shaped the city’s identity.

The city of Seville was the setting for one of the most important events in the personal biography of Emperor Charles V: his marriage to Princess Isabella of Portugal, which took place in the Alcazar on 11 March 1526. This marriage to his 23-year-old cousin, who could give him an heir, reconciled his economic needs as a Habsburg with the wishes of the Castilian Cortes of 1525 to Hispanicise the new monarch, who, born and educated abroad, appeared as an outsider in the eyes of his new Spanish subjects. In honour of the marriage of Emperor Charles V and Isabella of Portugal, a pavilion was built in the heart of the Alcazar [2]. The Pavilion of Charles V is an architectural jewel of Seville (Figure 1). Although its origins date back to the 12th century under Muslim rule, the marriage of Emperor Charles V and Isabella of Portugal led to its transformation into the Renaissance masterpiece we admire today. The Pavilion, ornamented with Genoese marble columns and intricate 16th century relief tiles (Figure 2), houses one of the most significant collections of Spanish Renaissance ceramics [3]. As a unique element of historical and architectural importance, it requires the application of advanced technologies for its conservation and understanding.

In the pursuit of preserving this architectural gem, we turn to the integration of Geomatics and ground penetrating-radar (GPR) techniques [4,5,6] and Historic Building Information Modelling (HBIM) techniques [7,8,9,10]. These methodologies separately provide useful results, but in conjunction [11,12,13] serve as indispensable tools for comprehensive documentation, providing meticulous insights into the Pavilion’s structural composition, materials, historical horizons [14,15,16], and ornamental intricacies. As we dig into safeguarding its physical integrity, integrating these techniques not only helps with a detailed structural evaluation [17,18,19] but also establishes a strong foundation for historical and archaeological research, enabling a deeper understanding of the pavilion’s evolution over time [5,8,9]. By leveraging HBIM examples and GPR applications within HBIM integration [20,21], we aim to underscore the significance of this integrative approach in the conservation and management of cultural heritage structures.

To reinforce the theoretical framework of our research, a review of relevant literature focusing on the integration of geometry and information into HBIM projects was made. Several key studies provide valuable insights into the methodologies and benefits of HBIM in managing historic structures. The HBIM-based management of built heritage is analysed in the work [22], using the case study of flooring and tiling in historic buildings, illustrating the practical applications of this technology. The authors of [23] explore the implementation and management of structural deformations within HBIM, showing its potential in preserving structural integrity and historical accuracy. The research [24] highlights the integration of artificial intelligence within HBIM to manage historic buildings, showcasing innovative approaches to enhance the accuracy and efficiency of heritage management. The study [25] examines the operability of point cloud data within an Architectural Heritage Information Model, emphasising the precision and reliability of data integration in HBIM projects. In addition, a rigorous HBIM project comprising 3D geometry, 2D documentation, and information of the Pavilion of Charles V is publicly available for exploration at Graphisoft’s BIMx Model Transfer in [26].

Through the synergy of 3D terrestrial laser scanner (3DTLS), global navigation satellite system (GNSS), and GPR [27], a precise and detailed point cloud of the pavilion and its subsurface is generated. This not only helps with the creation of building information modelling (BIM) of the pavilion but also enables researchers to explore the pavilion from diverse perspectives, fostering comparative analyses with contemporaneous structures.

Despite the absence of immediate structural problems, the collected data can be used to perform a detailed assessment to ensure the stability and longevity of the Charles V Pavilion. The structural safety assessment begins with a historical analysis to understand the original construction techniques, materials used, and earlier restoration efforts. GPR and geomatics techniques are used to inspect the internal structure and assess the effects of the water source within the building without causing damage. Subsequent visual inspections are conducted to identify any visible signs of distress, such as cracks and material degradation. The results of the material testing help to analyse the strength and durability of the construction materials. These results can be used to conduct a structural analysis using computational models, simulating the building’s behaviour under various loads and stress conditions.

Integrating these technologies goes beyond preservation; it opens avenues for the dissemination of cultural heritage [28,29]. Three-dimensional models offer immersive experiences for the public, allowing virtual exploration of every nook and cranny of the pavilion. Such models find applications in exhibitions, publications, and interactive platforms, extending the historical narrative to a wider audience.

To underscore the significance of such endeavours, recent GPR surveys at the Charles V Pavilion have unveiled potential archaeological remains, hinting at a historical timeline that could span 3000 years.

Our methods rely on integrating data using 3DTLS, GNSS, and GPR, processed through FARO Scene software. This results in a comprehensive BIM model of the Charles V Pavilion, streamlining coordination among teams, conflict detection, and precise documentation for various purposes. Drawing parallels with the Notre Dame Cathedral restoration project [30,31] in Paris, France, after the 2019 fire, showcases the effectiveness of BIM technology in preserving historical integrity and coordinating multidisciplinary efforts.

The integration of Geomatics, GPR, and BIM technologies in the study of the Charles V Pavilion stands as a beacon of innovation in cultural heritage preservation. Beyond safeguarding architectural wonders, it paves the way for global collaborations and advancements, contributing to the collective understanding and appreciation of our shared history. This paper digs into the intricate details of our interdisciplinary approach, highlighting the importance of advanced imaging techniques in the intricate realm of cultural heritage preservation.

## 2. Materials and Methods

The methodology used in this study aimed to conduct a comprehensive geophysical investigation of the Charles V Pavilion within the gardens of the Royal Alcazar in Seville, using non-destructive techniques, particularly GPR (Figure 3). The study was designed to achieve geometric characterisation and geophysical prospecting of the pavilion, providing reliable and objective documentation of its condition and layout. This documentation serves as a scientific reference for conservation projects, further research endeavours, and dissemination efforts.

The methodology was divided into two main phases:Data acquisition: GNSS observations were used to create the reference system. A detailed 3DTLS of the pavilion was conducted to capture its geometry, and concurrently, GPR surveys were carried out in the surrounding area to detect possible subsurface anomalies or structures.Integrated analysis: The obtained data were processed and co-registered to enable integration. Three-dimensional visualisation techniques and spatial analysis were employed to explore the relationship between surface and subsurface features, identify significant patterns and correlations, and generate hypotheses about the site’s evolution.

### 2.1. Georeferencing and System Projection

The project was referenced using the European Terrestrial Reference System 1989 (ETRS89), which serves as the official geodetic reference system in Spain. ETRS89 is part of the International Terrestrial Reference System (ITRS) and uses the GRS80 ellipsoid with specific parameters for geocentric origin and axis orientation. Observations were conducted using GNSS techniques, specifically using real-time kinematic (RTK) differential methods. The survey data were collected using a TOPCON Hyper II GNSS [32] receiver and processed using Topcon Tools [33] software to calculate coordinates in the ETRS89 system. For height determination, we used the geoid model aligned with the GRS80 geodetic reference system and referenced orthometric heights to the mean sea level in Alicante, ensuring accurate vertical positioning.

Once the reference frame was established using GNSS, the grid was staked out using a tape. This involved creating an exterior grid with 1 m intervals and an interior grid with 0.5 m intervals. The XYH coordinates obtained were used for staking out the grid, providing precise geospatial reference for the entire project.

### 2.2. 3D Digitisation through 3DTLS

The field campaign was carried out in December 2023; a FARO FOCUS X-330 [34] was used, and 32 scans were required. Calibrated spheres were used as tie points and the traverse checkerboards as references, which were adjusted with an accuracy of 2.7 mm for 95% of the points (Figure 4).

The initial step of the digitisation process involved the identification and establishment of control points strategically positioned around the pavilion. These control points served as reference markers for later scanning activities, ensuring accurate spatial registration and alignment of the acquired data.

During the scanning process, special attention was given to achieving optimal coverage and overlap to reduce data gaps and ensure continuity in the point cloud representation. The scanning operation was meticulously conducted from various vantage points surrounding the pavilion, allowing for comprehensive coverage of all exterior surfaces, including facades, columns, and decorative elements.

To enhance the accuracy and reliability of the digitised model, additional scanning passes were performed in areas of particular interest or complexity, such as ornate architectural details or areas with complex geometries. This targeted approach ensured the capture of fine-grained details essential for a comprehensive understanding of the pavilion’s architectural morphology for the later BIM modelling.

After completion of the scanning process, the collected point cloud data underwent rigorous quality control procedures to identify and rectify any anomalies or inconsistencies. Noise and artifacts were removed, and the data were further refined through filtering and optimisation techniques to enhance clarity and precision [35,36].

The processed point cloud data were transformed into a unified three-dimensional digital model using FARO Scene v. 2019 Software (FARO Technologies Inc., Lake Mary, FL, USA). The block adjustment technique [37] was used to refine the spatial alignment of data points, ensuring consistency and accuracy throughout the dataset. This digital model accurately depicted the pavilion’s geometry and morphology, helping with detailed analysis and visualisation of its architectural elements and spatial configuration.

This digital model serves as a valuable resource for architectural documentation, heritage conservation, and research endeavours, enabling stakeholders to explore and analyse the pavilion’s rich architectural heritage with unparalleled depth and clarity.

### 2.3. Subsurface Digitisation through GPR

The GPR method is generally more effective than other geophysical methods for surveying underground structures. GPR has become increasingly important as a common technology for the detection of structures and artefacts in archaeological and cultural heritage surveys. This tool is often more effective in the study of underground structures than other geophysical methods [38,39,40]. It provides accurate results and ensures complete site coverage. This technique is also a valuable cartographic tool due to its capability for high-resolution subsurface imaging and 3D data representation [41,42,43,44,45]. GPR is a non-destructive technique that emits short-duration electromagnetic pulses in the UHF-VHF frequency range and relies on the reflection of these pulses to detect electromagnetic variations in subsurface materials [46,47].

Data collection was required for the whole pavilion under study. Thus, three-dimensional GPR methodologies were performed to identify and define subsurface structures and anomalies within the vicinity of the Charles V Pavilion. The GPR system used was the GSSI SIR4000 with a Utilityscan cart, equipped with dual-frequency antennas (300/800 MHz), ideal for both shallow and deeper penetration depths up to 2–3 m. The chosen frequencies ensured versatility in detecting subsurface features while maintaining high resolution and sensitivity to subtle variations in electromagnetic properties [48,49].

Before data collection, the survey area was systematically gridded each 0.5 m to help with systematic coverage and ensure uniform sampling of the subsurface (Figure 5). Transects were carefully planned based on the site’s historical significance, architectural layout, and suspected areas of interest identified through prior research and consultation with archaeological experts.

### 2.4. GPR Model to BIM Integration

In the context of 3D GPR analysis, the acquired radargrams undergo a critical transformation process to help with integration into BIM workflows [50,51,52] (Figure 6). In the workflow for integrating 3D GPR data into BIM environments, these sequential steps are undertaken:Slicing: Initially, the 3D radargrams are sliced into 2D representations with a 3 cm equidistance and thickness to establish accurate depth profiles.Merging: Multiple slices are combined into a single file, new output dataset using a Merge tool, ensuring comprehensive data integration.Reclassify: The values within the raster data are reclassified by using a threshold to be or not anomaly and in this way to enhance accuracy and relevance in representation.Conversion to polygon: The reclassified raster datasets are converted into polygon features, facilitating easier interpretation and visualisation within the BIM environment.CAD exportation: CAD drawings based on the converted polygon features are created using the Export to CAD tool, providing a format compatible with CAD software.Extrusion: The exported CAD drawings undergo extrusion to convert 2D representations into 3D models, capturing the spatial relationships of subsurface features with architectural elements.Integration to the BIM model: Finally, the extruded 3D models are integrated into the BIM environment, enriching the model with detailed information about the subsurface features of the cultural heritage site, such as the Charles V Pavilion.

The aim of this integrated approach is to enhance the understanding of the site and aid in informed decision making for conservation and research strategies within the BIM framework.

### 2.5. Data Processing and Analysis

The processing and analysis of the collected data from both the 3DTLS and GPR surveys were essential in extracting meaningful insights into the Charles V Pavilion and its surroundings.

For the 3DTLS data, a systematic workflow was used to process the raw point cloud data obtained from the scanning process. Pre-processing steps, including noise removal, outlier detection, and registration, were conducted to ensure data integrity and accuracy. Feature extraction and segmentation algorithms, such as random sample consensus (RANSAC) [53,54] and region growing [55], were then applied to identify key architectural elements and ornamental details. Algorithms for surface reconstruction and mesh generation helped with the creation of detailed 3D models of both the exterior and interior spaces of the pavilion.

Initially, a high-resolution scan was conducted using a laser scanner to capture detailed geometry of the pavilion in the form of a three-dimensional point cloud. This point cloud provided an accurate representation of the pavilion’s surface and served as the starting point for subsequent modelling in computer-aided design (CAD) software such as Revit.

Subsequently, specialised software was employed to process the point cloud and generate a three-dimensional mesh model. This process involved the application of advanced mesh reconstruction algorithms to convert the point cloud data into a structured representation of the pavilion’s surface. Once the mesh modelling was completed, the resulting model was imported into Revit for further refinement and adjustment.

In Revit, additional modelling tasks were carried out to incorporate specific architectural details and enhance the pavilion’s representation. This included adding structural elements, decorative features, and other relevant details to achieve an accurate and detailed depiction of the building.

After the acquisition of the field data, the GPR data underwent processing steps to improve the clarity and interpretability of the radar profiles. Post-acquisition processing procedures must be applied to the raw GPR field data set before a 3D image with all reflection profiles (GPR records) in a grid can be produced. Due to good signal penetration, a basic data processing procedure was applied to the raw data set using RADAN 7 software developed by Geophysical Survey Systems, Inc. (GSSI). As a first step, zero-time correction, background removal and gain function were applied to amplify the received signal and improve the identifications of reflections. The 2D data were then processed by applying filters such as the Kirchhoff migration filter using the average velocity for diffraction removal. To compute the average velocity of the EM wave propagation in the subsurface of the pavilion, we proceeded to calculate the average velocity of the GPR wave using the hyperbolas fitting method on a set of hyperbolas recorded in different profiles, obtaining an average velocity value of 0.0874 cm/ns. The dielectric permittivity (*ε*) was calculated to be 11.76, according to the following equation [46,47]:(1)$$\varepsilon ={\left(\frac{c}{v}\right)}^{2}={\left(\frac{ct}{2h}\right)}^{2}$$
where *h* is the depth, *t* is the two-way travel time, *v* is the electromagnetic wave velocity, and *c* is the velocity of light in free space (*c* ≈ 0.2997 cm/s).

As a result, a time-to-depth conversion was performed for each processed reflector profile in the investigated pavilion subsurface.

Next, a GPR-3D model of the subsurface was obtained by aligning the processed 2D profiles to accurately locate burial structures and pavement evidence beneath the studied sector. Depth-slice amplitude maps of the 3D model were used to identify features at a constant depth. Once the depth-slice maps were produced, the dataset was visualised using the isoamplitude surface rendering technique (isosurface). In addition, a transparent visualisation of the 3D GPR dataset was carried out to improve the visibility of subsurface anomalies and features.

As previously commented, integrating 3DTLS and GPR data was achieved through a process ensuring spatial coherence and compatibility. First, the reference system for both datasets was established using GNSS technology, guaranteeing consistent spatial alignment. Then, each dataset was adapted to this common reference system, ensuring unified integration and correlation between the 3DTLS point cloud data and GPR subsurface scans.

In the processing pipeline, the results from FARO Scene for 3DTLS were exported to Autodesk Revit for detailed modelling and analysis. Leveraging the capabilities of Revit, the 3DTLS data was transformed into accurate 3D models of the study area’s architectural elements. Meanwhile, the GPR anomalies, processed using RADAN software, underwent a dedicated workflow to create 3D Extruded CAD Anomalies. This approach helped with a comprehensive understanding of both surface and subsurface features within the study area.

For quantitative analysis, many methodologies were used to assess dimensions, volumes, and spatial distributions within the HBIM System. This encompassed conducting volumetric calculations and statistical measurements using software packages such as Autodesk Revit. Leveraging the integrated TLS derived 3D model and GPR data, dimensional analyses were conducted to precisely measure structural elements like walls, columns, and beams, while in future volumetric analyses, they will allow for the accurate calculations of spaces and material quantities. Spatial distribution analyses were performed to visualise the distribution of features such as cracks, voids, and material properties, providing insights into structural integrity and degradation patterns. Additionally, condition assessments can be carried out to map and analyse cracks and material degradation, guiding conservation efforts. These quantitative analyses not only enhance the understanding of the heritage structure’s condition but also inform decision making for preservation and restoration strategies.

Visualisation techniques played an essential role in communicating findings effectively and enabling immersive exploration. Data were exported to advanced rendering capabilities software such as Blender to create 3D visually compelling representations renderings of the study area. Additionally, data were also imported to Unity, and they are ready to be exported to virtual reality devices to give stakeholders immersive experiences, allowing them to interactively explore existing architectural and historical subsurface features. The data processing and analysis workflow yielded valuable insights into the architectural, archaeological, and geological parts of the Charles V Pavilion and its surroundings. This integrated approach permits a deeper understanding of the pavilion’s historical significance and informs conservation and research strategies for this cultural heritage site. The methodology used served as a robust framework for scientific inquiry and conservation efforts, providing valuable insights into the architectural and geological heritage of the Charles V Pavilion in Seville.

### 2.6. 3D Digitisation of the Charles V Pavilion

The 3DTLS survey of the Charles V Pavilion yielded a complete point cloud dataset capturing detailed geometric information about the structure’s exterior and interior. The resulting 3D models represented the Pavilion’s architectural features, including complex ornaments, columns, and facades. The digital documentation helped with virtual exploration and visualisation of the pavilion from various viewpoints, enhancing our understanding of its spatial layout and aesthetic features.

The point cloud was exported in RCP format to Autodesk ReCap software [56] (Autodesk, San Rafael, CA, USA) to understand and verify existing conditions and on-site elements for further insight and better decision making. Subsequently, it was exported to Autodesk Revit software [57] (Autodesk, San Rafael, CA, USA) in RCS format and worked on as a complete 3D modelling object within the Revit interface. The point information was manipulated to be displayed in various modelling views (Figure 7): floor plans, elevations, sections and in 3D view. A floor structure was created to accurately position the elements in the model: Level 1 (pavement), Level 2 (both cornices), and Level 3 (dome).

### 2.7. GPR Survey

The 2D reflection profiles provide an overview of the surveyed area of the archaeological site. These radar profiles illustrate the subsurface dissimilarities. In addition, the profiles show main anomalies at depths ranging from 0 to 1.10 m (cultural layer 1), from 1.10 to 1.95 m (cultural layer 2), from 1.95 to 2.80 m (cultural layer 3), and from 2.80 to 3.70 m (cultural layer 4), as shown in (Figure 8a). However, determining the geometric and dimensional features of structures from 2D GPR data is time-consuming and requires individual radar profile analysis. The 3D GPR visualisation techniques overcome this drawback by characterising the buried remains for the whole studied subsurface volume (Figure 8b). These anomalies reveal the presence of several cultural layers, which may have buried remains, on which the Charles V Pavilion was built. In addition, the isosurface rendering technique enabled the visualisation of surfaces of equal amplitude in the studied volume. This 3D GPR data visualisation reproduced and highlighted in particular the geometric features of the reflections, derived from archaeological remains, with a strong contrast with the surrounding environment of the analysed volume. This isosurface image was coloured in grey-black to show these amplitude values, while others were made transparent to better represent only the archaeological remains and simplify their detection and data interpretation. Transparent visualisation of the 3D GPR dataset was carried out to reveal in 3D the main anomalies (buried structures) in the studied volume (Figure 8c). These results shed light on the historic building evolution of the area where the pavilion is located.

## 3. Results

### 3.1. Integration of 3DTLS and GPR Data

Integration of 3DTLS and GPR data enabled a holistic analysis [58] of the pavilion’s surface and subsurface features. Co-registration of 3DTLS-derived 3D models with GPR depth slices helped with the correlation of surface features with subsurface anomalies, enhancing our understanding of the pavilion’s architectural layout and subterranean structures. The integrated dataset provided valuable insights into the pavilion’s historical significance and archaeological potential, guiding future research and conservation efforts.

Despite being a prominent example of Spanish architecture and a significant historical monument, its relationship with the surrounding subsurface has been understudied. This study addresses this knowledge gap by applying advanced data acquisition and analysis techniques to better understand the interaction between the pavilion’s architecture and the geological and archaeological features of the subsurface.

As for the accuracy of our work, each technology used in this project offers different levels of precision and resolution, which are critical for ensuring reliable analyses. The TOPCON Hyper II GNSS in RTK mode used provides horizontal accuracy typically within 1–2 cm and vertical accuracy within 2–3 cm. Each point was measured 10 times to average the result and try to reach centimetric accuracy in the reference frame vertices observations. The 3D TLS conducted using the FARO X-330 provides millimetre-level accuracy, generally around 2 mm in range determination of a single point in this context, but the scans were used to fit known radius spheres and adjust them as a block according to [37] to obtain a spatial alignment accuracy of 2.7 mm for 95% of the points. GPR scans, conducted using the GSSI DF 300/800 utility scan, offer a vertical resolution that depends on the frequency of the antenna and the dielectric properties of the material being surveyed. For a 300 MHz GPR antenna, the vertical resolution is typically approximately 10 cm in our case study. In contrast, an 800 MHz GPR antenna offers a finer vertical resolution, typically around 3 cm in our material properties.

GPR data can be visualised in various formats to enhance interpretation. One common representation is as a raster layer, where each pixel corresponds to a specific location and has the amplitude of the radar signal reflected from the subsurface (Figure 9). This raster representation provides detailed information about the strength and distribution of subsurface reflections.

GPR data can also be displayed as a vector layer, where thresholds are applied to binarise the data (Figure 10). In this format, areas exceeding a certain signal strength threshold are delineated, allowing for the identification and mapping of subsurface anomalies with greater clarity. The availability of both raster and vector representations offers flexibility in analysing GPR data, catering to different research objectives and enhancing the overall understanding of subsurface features beneath the Charles V Pavilion and its surroundings.

All the processed GPR data were in one way or another then interpreted to identify and characterise subsurface features of archaeological significance.

### 3.2. Quantitative Analysis and Visualisation

Integrating Revit models with subsurface data enabled a comprehensive quantitative analysis and visualisation of the architectural elements, subsurface anomalies, and archaeological features within the study area. Leveraging the capabilities of Revit, architectural elements were accurately modelled and measured, providing precise metrics for assessing their dimensions and spatial distribution. Simultaneously, subsurface anomalies detected through GPR were incorporated into the analysis, allowing for the characterisation of underground features (Figure 11).

## 4. Conclusions

The results of the integrated imaging and geophysical survey of the Charles V Pavilion provide valuable insights into its architectural, archaeological, and historical significance.

Integrating 3DTLS and GPR data offers many advantages, particularly in the context of our study on the Charles V Pavilion. While obtaining visually appealing representations has been a beneficial outcome of our integrated data, our primary focus has been on harnessing the comprehensive insights gained through data integration and analysis. Firstly, integrating 3DTLS and GPR data allows for a complete understanding of both the surface and subsurface features of the pavilion and its surroundings. By combining high-resolution 3D models generated from 3DTLS data with detailed subsurface imaging from GPR, we can create a unified building information model (BIM). This BIM serves as a valuable asset, giving stakeholders a comprehensive digital representation of the pavilion that incorporates both architectural features and subsurface anomalies. Furthermore, integrating 3DTLS and GPR data enables enhanced spatial analysis and visualisation capabilities. By overlaying the 3D model of the pavilion with GPR anomalies, we can identify correlations between surface features and subsurface anomalies, helping with a deeper understanding of the structural integrity and historical context of the pavilion. This integrated approach not only enhances our ability to assess the condition of the pavilion but also enables us to make informed decisions about preservation and restoration efforts.

Through this interdisciplinary methodology, we have unlocked new insights into the complex relationship between surface structures and subsurface features, enriching our understanding of historical landscapes and archaeological contexts [58,59].

Simultaneously, incorporating subsurface data, acquired through ground-penetrating radar surveys, provided a deeper understanding of the site’s underlying geology and archaeology. Interpretation involved the identification of reflection patterns, anomalies, and discontinuities indicative of buried structures, archaeological remains, or geological formations.

The identification of subsurface anomalies, such as buried foundations or infrastructure, offered valuable clues about past land use and human activity, contributing to our knowledge of historical development patterns.

Furthermore, integrating Revit models and subsurface data can enable quantitative analysis and statistical assessments, revealing patterns and correlations within the dataset. These findings not only enhance our understanding of individual architectural features but also inform broader interpretations of site evolution and cultural significance (Figure 12).

The findings of this study have important implications for understanding the history and evolution of the Charles V Pavilion, as well as for the conservation and management of cultural heritage. Integrating architectural and geophysical data provides a multidisciplinary perspective, highlighting the importance of considering the subsurface in the planning and preservation of historical heritage.

The structural safety data could be included in Revit for numerical modelling. The software can incorporate material properties of the pavilion to precise geometric representations of the pavilion by integrating data from geomatics techniques, such as 3DTLS and photogrammetry, to reflect the building’s current state with high fidelity. In future, this model can be exported to structural analysis software, helping with complex simulations to evaluate the pavilion’s response to various loads and stress conditions. Additionally, finite element analysis (FEA) can be conducted to assess stress distribution, identify potential failure points, and test reinforcement strategies. By simulating different restoration interventions, Revit aids in evaluating their impact on structural safety and makes sure proposed measures align with historical preservation standards.

With artificial intelligence (AI) and deep learning algorithms [60,61], data fusion techniques can significantly enhance the depth and accuracy of analysis in the near future. By leveraging AI and deep learning algorithms, such as convolutional neural networks (CNNs) and recurrent neural networks (RNNs), data fusion can extract complex patterns and relationships from heterogeneous datasets, including those derived from 3DTLS and GPR surveys. These advanced algorithms can integrate multi-modal data sources, such as point clouds, radar profiles, and visual imagery, enabling comprehensive analysis of the Charles V Pavilion and its surroundings. Furthermore, AI-driven data fusion can automate feature extraction, anomaly detection, and classification tasks, allowing for efficient and accurate interpretation of complex geospatial datasets. By combining the power of data fusion with AI and deep learning, researchers could uncover deeper insights into the architectural, archaeological, and geological aspects of the study area, thus advancing our understanding of the historical significance and conservation needs of cultural heritage sites like the Charles V Pavilion.

The models developed in our research will be exported to Unity to facilitate immersive visualisation through virtual reality. This approach will enable stakeholders to interactively explore the Charles V Pavilion and its surrounding historical contexts in a highly engaging and intuitive manner. By integrating data, such as detailed architectural features, material properties, and subsurface anomalies detected through GPR, the VR environment will provide a comprehensive and multi-layered understanding of the site. This immersive experience aims to enhance the analysis, presentation, and decision-making processes related to the preservation and restoration efforts, offering a powerful tool for both researchers and conservationists.

The detailed 3D models and subsurface imaging data offer a comprehensive understanding of the pavilion’s spatial layout, construction history, and cultural context. These findings contribute to the preservation, research, and interpretation of this iconic cultural heritage site, guiding future conservation efforts and helping with public appreciation and engagement with the pavilion’s rich heritage.

## Figures and Tables

**Figure 1 jimaging-10-00128-f001:**
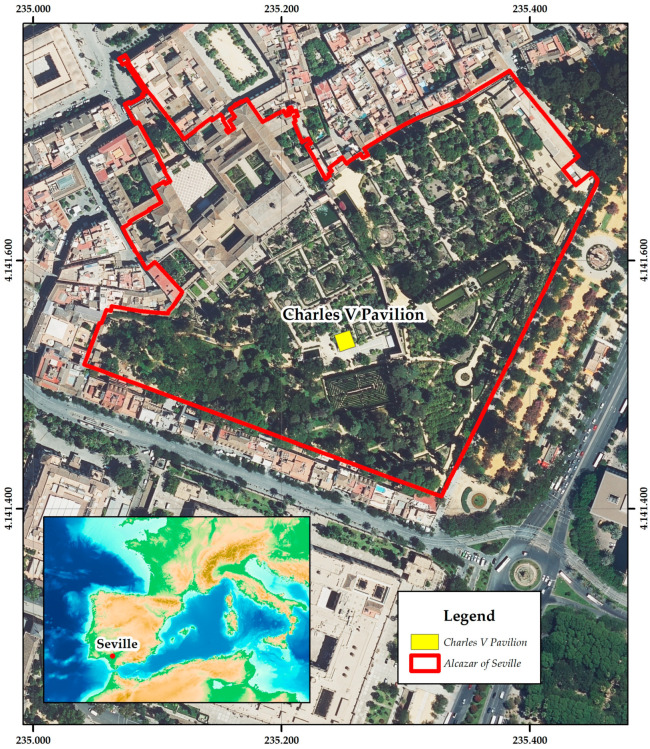
Location of the Charles V Pavilion in the Alcazar of Seville.

**Figure 2 jimaging-10-00128-f002:**
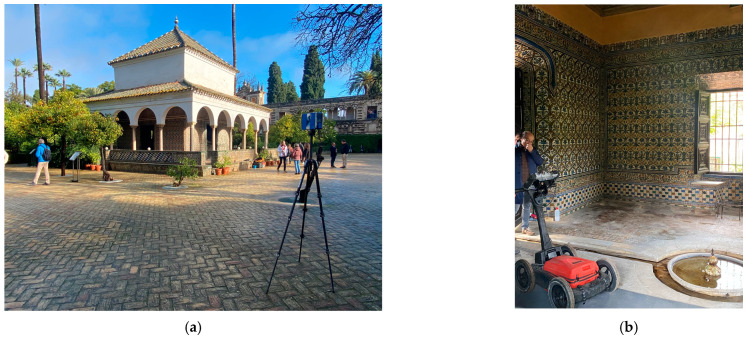
The Charles V Pavilion in the gardens of the Royal Alcazar of Seville with the utilised technology. (**a**) Exterior view with the 3DTLS. (**b**) Interior view with the GPR.

**Figure 3 jimaging-10-00128-f003:**
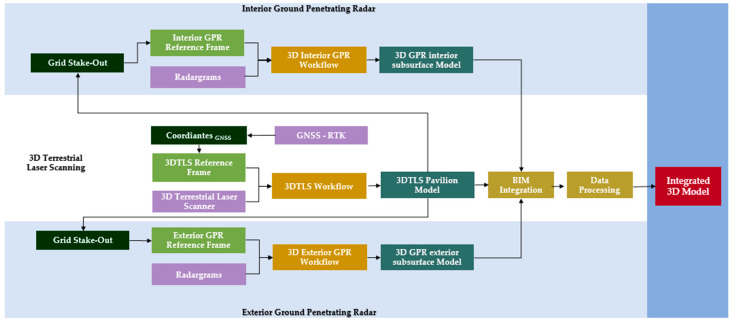
General workflow.

**Figure 4 jimaging-10-00128-f004:**
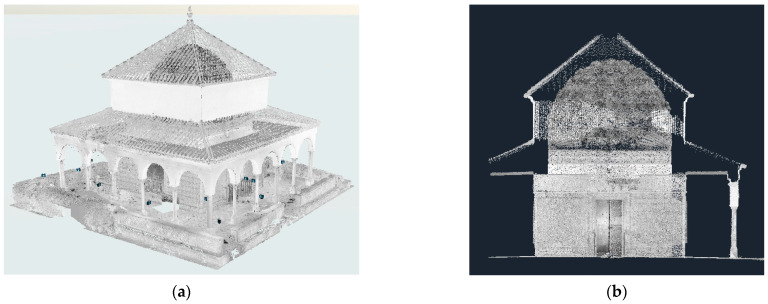
(**a**) Point cloud of the adjusted block and (**b**) cross-section of the adjusted point cloud.

**Figure 5 jimaging-10-00128-f005:**
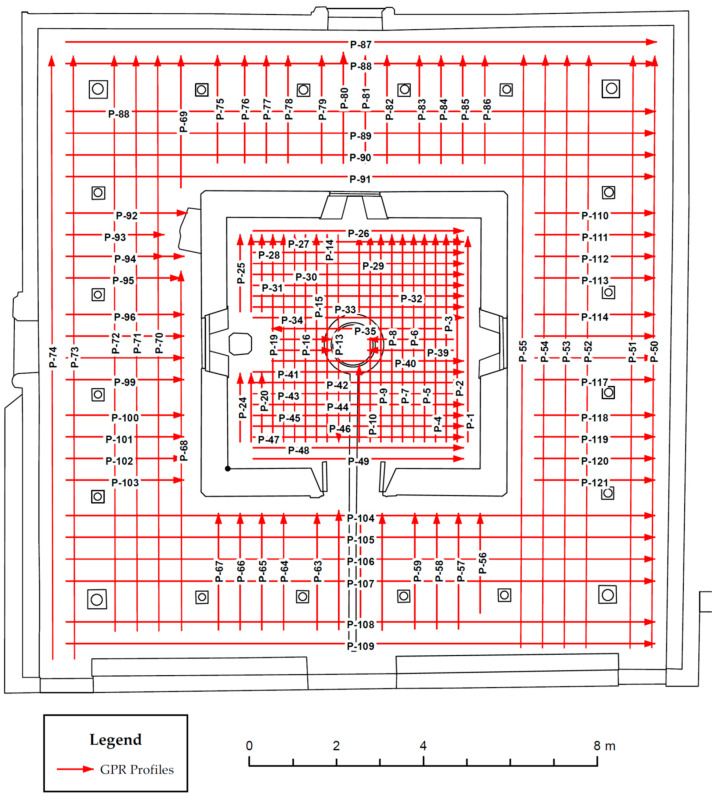
Created grid for the GPR study.

**Figure 6 jimaging-10-00128-f006:**
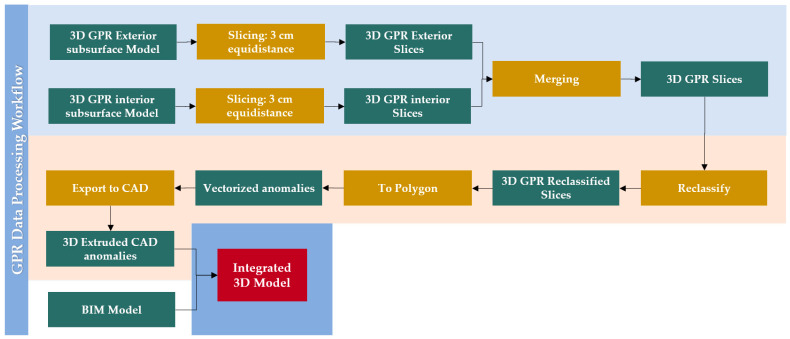
General workflow to facilitate GPR integration to BIM.

**Figure 7 jimaging-10-00128-f007:**
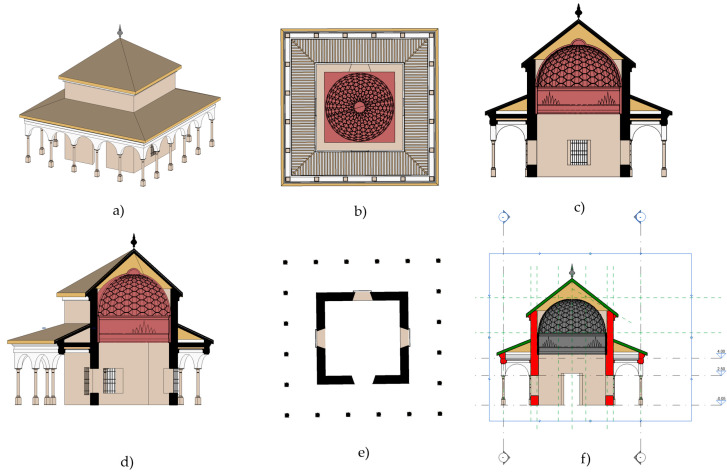
Views of the Pavilion of Charles V BIM model: (**a**) 3D view, (**b**) bottom view, (**c**) orthographic section of the 3D model, (**d**) section of the 3D model, (**e**) plant plane, and (**f**) horizontal and vertical section positions.

**Figure 8 jimaging-10-00128-f008:**
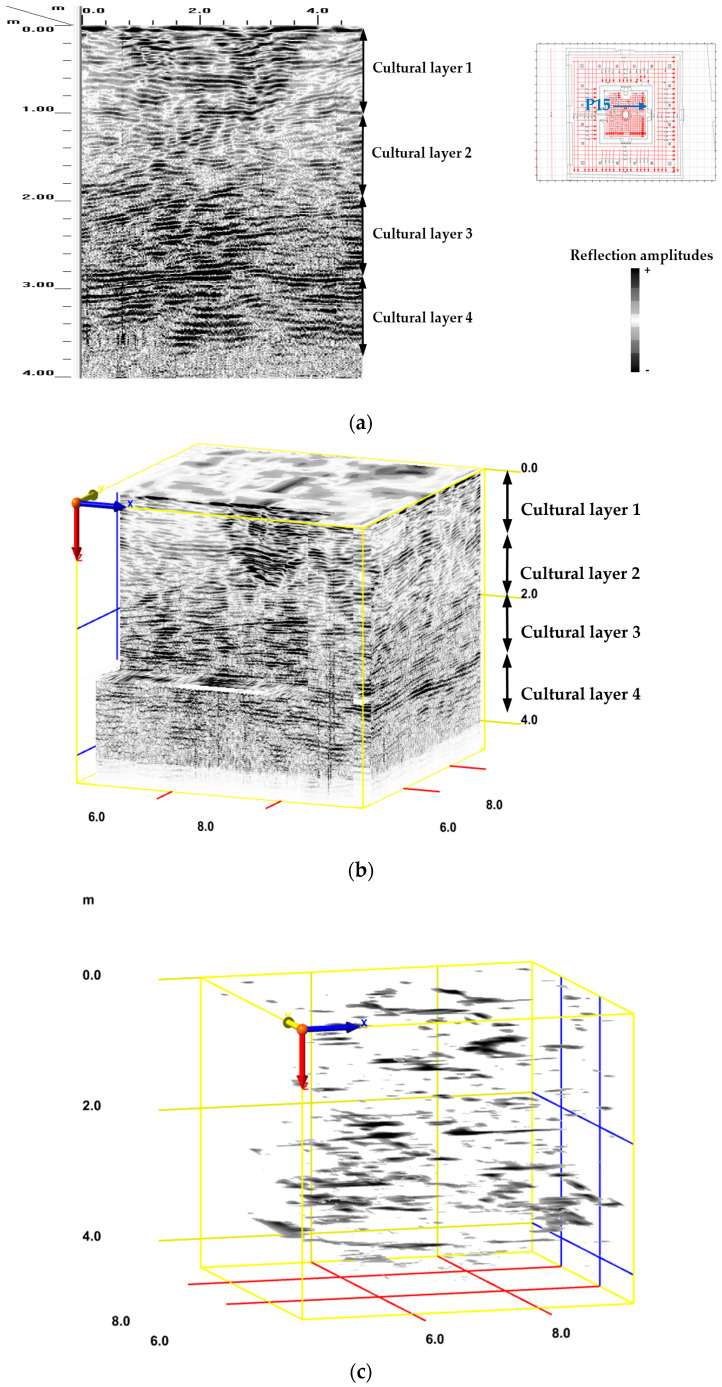
(**a**) P15 reflection profile and (**b**) 3D radar section images after processing the raw data, showing underground evidence of structures and cultural layers; their layer thicknesses are indicated by black arrows. (**c**) Isosurface image in the studied volume, showing the main anomalies (buried structures) detected and other reflection points coloured in grey-black.

**Figure 9 jimaging-10-00128-f009:**
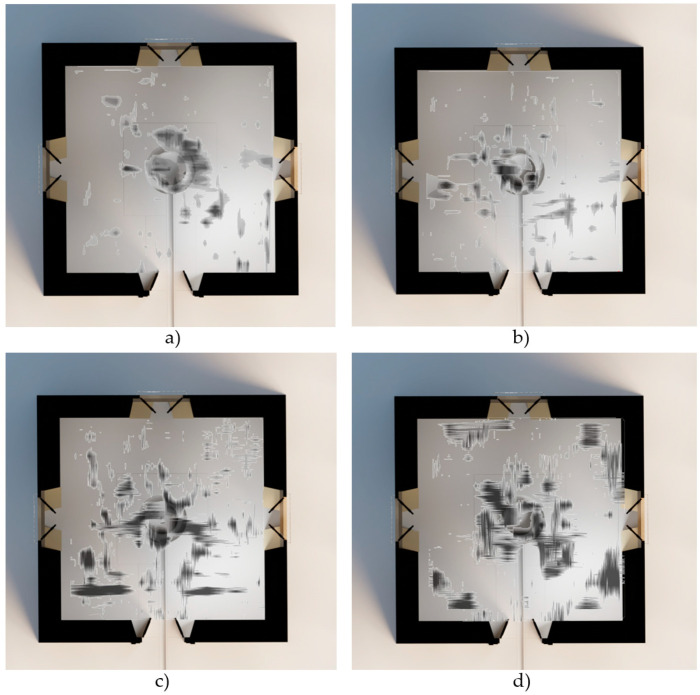
Examples of overlapping raster anomalies in the 3D model due to the different structures detected in each cultural layer: (**a**) 66 cm depth in cultural layer 1, (**b**) 133 cm depth in cultural layer 2, (**c**) 222 cm depth in cultural layer 3, and (**d**) 311 cm depth in cultural layer 4.

**Figure 10 jimaging-10-00128-f010:**
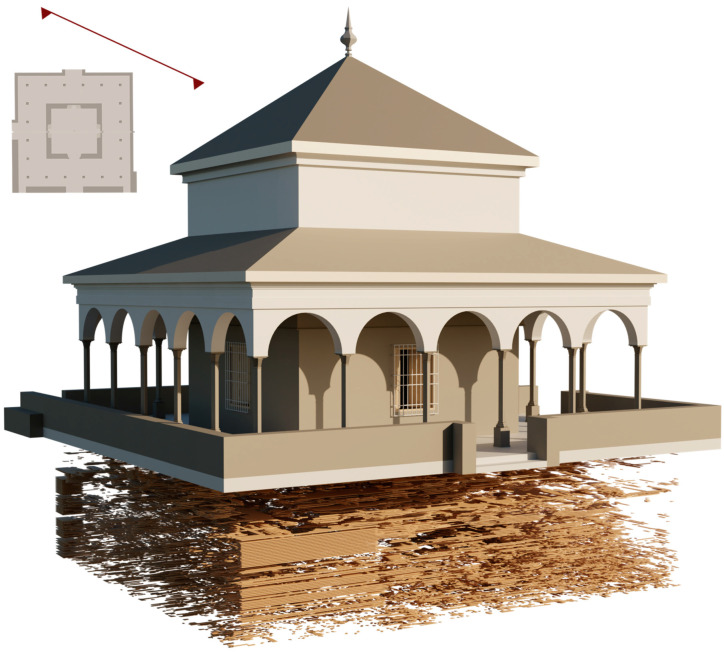
Integration of GPR data into the 3D model.

**Figure 11 jimaging-10-00128-f011:**
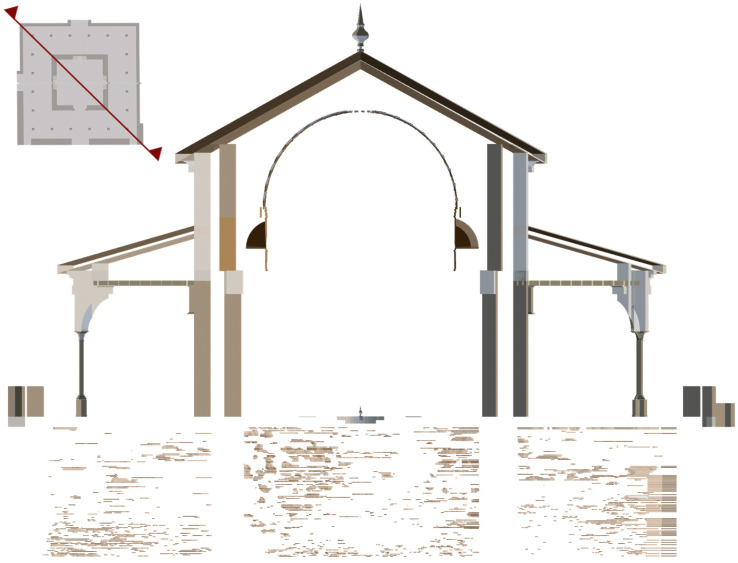
Cross-section of the pavilion.

**Figure 12 jimaging-10-00128-f012:**
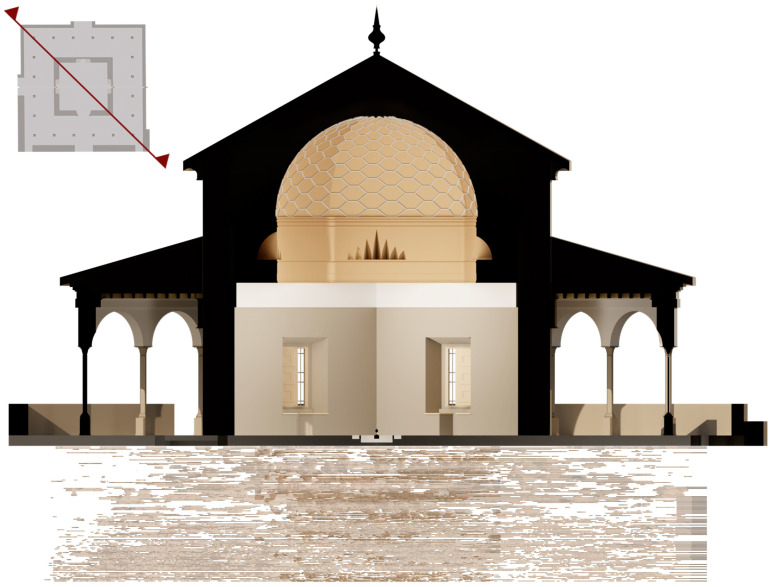
Combined interior view of the integration.

## Data Availability

The research data supporting this publication are not publicly available. The data were collected by GIM Geomatics and are kept by World Monuments Fund.

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
