# Peer review of "Integrated Building Modelling Using Geomatics and GPR Techniques for Cultural Heritage Preservation: A Case Study of the Charles V Pavilion in Seville (Spain)"

_2313-433X, 2024, doi:10.3390/jimaging10060128_

Round 1
Reviewer 1 Report
Comments and Suggestions for Authors
This manuscript reports about an integrated strategy for preservation of CH buildings. The discussed case study is the Charles V Pavilion in Seville (Spain). Terrestrial Laser Scanner, Global Navigation Satellite System, and Ground Penetrating Radar are employed to build a Building Information Model that can be employed by stakeholders for preservation and decision-making.
It is opinion of this referee that the manuscript can be considered for publication provided that following suggestion will be considered by the authors:
1) While the discussion of the methodology is clearly discussed by the authors, the state-of-the-art is not properly represented.
A lot of recent studies discuss the preservation strategy proposed by the authors in the framework of the Heritage-BIM. The authors can consider, for instance, following studies:
- De Falco A., et at. An HBIM Approach for Structural Diagnosis and Intervention Design in Heritage Constructions: The Case of the Certosa di Pisa. https://doi.org/10.3390/heritage7040088
- Ciuffreda A.L., et al. Historic Building Information Modeling for Conservation and Maintenance: San Niccolo’s Tower Gate, Florence. https://doi.org/10.3390/heritage7030064
- Monchetti S., et al. Insight on HBIM for Conservation of Cultural Heritage: The Galleria dell’Accademia di Firenze. https://doi.org/10.3390/heritage6110363
- Miccinesi L., et al. Ground Penetrating Radar Survey of the Floor of the Accademia Gallery (Florence, Italy). https://doi.org/10.3390/rs13071273
2) The goal of integrating GPR data within the BIM should be better specified through the manuscript. It should also specified as the recording (and utility) of this information can be generalized for other CH buildings.
3) Some information must be provided about the structural safety assessment of this structure. In this respect the possibility to use the Revit model to build a numerical model of the Charles V Pavilion, should be discussed.
Comments on the Quality of English Language
Minor editing of English language is needed
Author Response
Dear Reviewer,
We would like to really thank you for your detailed review of our manuscript. We greatly appreciate the time and effort you have invested in providing valuable feedback. We want to assure you we have carefully considered all of your comments and suggestions. To ensure transparency, we have prepared a detailed point-by-point response to address each of your concerns. We invite you to review the attached PDF document, where you will find our responses elaborated upon. Your feedback has helped to improve the quality and clarity of our manuscript, and we are grateful for your contributions to the scholarly dialogue.

Reviewer 2 Report
Comments and Suggestions for Authors
Dear Authors,
I read your manuscript thoroughly, and I appreciate the work you have done. The topic is very interesting and aligns with the world's trends in combining GPR with other spatial data. Here, you have also used GPS and TLS, which is convincingly justified and clearly explained. Nevertheless, while reading the text, I came across some unclarities needing explanation.
First, as I can see, you cite almost exclusively Spanish authors. I understand they are recognized specialists in this area; however, scientific writing should be universal. There are numerous similar publications worldwide, so please undergo a solid literature review and extend your list of authors (examples).
Secondly, please structure the geodetic/cartographic items. Lines 102-108 contain information about the geodetic reference system ETRS89. It's the European part of ITRS - in short - it's not a cartographic projection but an international reference system based on GRS80 ellipsoid, etc. For more details, please refer to any geodetic book (e.g., about Higher Geodesy, GNSS, or Satellite Geodesy). However, please provide the readers with short information about the XYH coordinates used as well as the method of staking out your grid. Did you use any geoid model for determining heights, or maybe - indeed - you use global coordinates expressed in ETRS89? For the surveyors reading your text (many of them will do so because they deal with similar things worldwide!), this passage is unclear.
I couldn't find any information about the accuracy of your work (except for line 377 about TLS). Each technology you mention offers slightly different accuracy and procession (surveying resolution). This should be discussed before further analyses are conducted.
The passage starting with Line 175 until 190: I don't understand your intention of giving this parameter, but - anyway - if you already do it, please explain your motivation. In my opinion, the sentence: "This value was applied for calculation of the processed depths in the investigated Pavilion" (line 189) tells us nothing because that's the way GPR works. Is it you who calculated the depths or you had rather done it in the dedicated software? The passage ends suddenly without any ties to the following section.
Line 237 - please check the whole text and eliminate such minor typos.
Lines 249-256: the data integration is vaguely explained. Did you do it with a special software, or maybe it is your own finding? Please provide the readers with more details. What is more, the qualitative analyses should also be explained in detail (which statistical tests? How did you do it? The current passage is far too general).
Finally - what is the overall advantage of integrating TLS and GPR in your case? I know, your intention is to write about that, but paradoxically, you only present a nice picture of the pavilion with some underground detection echogram. Did you model it somehow and expressed it together in a common building information model? What kind of asset can it give? What is the convincing novelty in your presented approach? I know the answers and I am sure you know them too! Regarding that - please provide them convincingly!
I wish you good luck with your work!
Comments on the Quality of English LanguageThe English language is correct and grammatically appropriate. I do not see any significant mistakes. On the contrary, the text is written clearly and kept as simple as possible. Some minor typos were identified, such as a lack of spaces in acronyms (like 3DTLS, which should be 3D TLS), etc. I recommend providing a final proofreading before publishing the text.
Author Response

(The authors gave the same response as above.)

Reviewer 3 Report
Comments and Suggestions for Authors
This paper is intended to delve into the above-ground level and sub-sufrace geometries of the 'Pabellón de Carlos V' (Pavilion of Charles V) in the 'Jardín de la Alcoba' (Alcoba Gardens), located in the 'Real Alcázar de Sevilla' (Seville, Spain). Authors worked on this by conducting both, GNSS-georeferenced 3D Terrestrial Laser Scanning and Ground Penetrating Radar surveys, which are advanced qualitative and quantitative 3D recording technologies. To be highlighted is the thorough GPR layout planned and its developed survey. Research into the physical and geophysical characteristics of historic buildings and sites are always welcome.
However, this manuscript shows important deficiencies in:
- Structure, as content should be condensed into the corresponding sections;
- Methodological processes, as further details are needed to rigurously describe the approaches and processes;
- Integrated outcomes, as there is no evidence on the linking of quantitative data from the GPR survey to 3D geometries into the BIM project. That is the real decision-making trigger this research should pursue.
That lack of evidence on the integration of data from quantitative analysis into the HBIM developed may indicate that 3D geometry from third-party software was merely imported into Revit to 'attach' the sub-surface 3D model to the 3D model of the building. The aimed BIM integration should show how and that the quantitative data obtained were linked to BIM geometries.
For their part, there is no evidence on neither the data sharing nor the augmented/virtual reality visualisation the Authors claim to have achieved in the Abstract. One also expects to find this in the manuscript.
Some specific comments:
INTRODUCTION
The first paragraph may lead the Readers to think that the 'Catedral de Santa María de la Sede' (Cathedral of Seville) and the 'Archivo de Indias' belong to the Reales Alcázares. Please rewrite it so that it is clearly understood that they are independent, although the three of them constitute the monumental complex listed by UNESCO. Citing the source is also recommended:
[1] United Nations Educational Scientific and Cultural Organization (UNESCO), Cathedral, Alcázar and Archivo de Indias in Seville, (1987) 383bis. http://whc.unesco.org/es/list/383#top (accessed November 7, 2016).
Lines 67-68: "the creation of Building Information 67 Modelling (BIM)". BIM is the technology or methodology, and is already established. What can be "created" is a BIM of a case study (a Building Information Model) or the BIM project of it. Please refer to the correct expression: BIM technology/methodology or the BIM project.
Finally, mainly given their substantial number of citations, and to reinforce the literature review of this manuscript, there exist research publications that should not be ignored on the integration of both geometry and information into the HBIM of the specific case study of the Pavilion of Charles V of the Reales Alcázares de Sevilla. In this sense, a basic search on search engines such as Google Scholar, Scopus, or WoS yields publications on the creation of such HBIM projects, the evaluation of the 3D modelling accuracy, or the identification and classification of the building's structural deformations. Please do not forget that science should contribute to existing knowledge, so a strong theoretical framework is mandatory.
In addition, a rigurous HBIM project comprising 3D geometry, 2D documentation, and information of the Pavilion of Charles V is publicly available for exploration at Graphisoft's BIMx Model Transfer at:
https://bimx.graphisoft.com/model/e7bbf092-341e-4470-bf0b-d514b32c844b
MATERIALS AND METHODS
2.1. GNSS
GNSS positioning accuracy should be specified.
2.2. TLS
3D survey accuracy (TLS device settings) should be specified, understood different from the 2.7 mm accuracy after registration.
Please describe the "detailed analysis and visualization of its architectural elements and spatial configuration" conducted, if any.
2.5. In which way did 3D surface and mesh reconstruction helped creating interior and exterior 3D models of the Pavilion? Please describe in detail.
Please describe the 'TLS point cloud data - GPR data alignment' process (lines 249-252).
Lines 253-257: "Quantitative analysis, including volumetric calculations and statistical measurements, was conducted to assess the dimensions, volumes, and spatial distribution of architectural elements and subsurface features through the BIM System. Visualization techniques, including 3D rendering and virtual reality (VR) were used to effectively communicate findings and enable immersive exploration.". As indicated in the beginning of this review, there is no evidence on all this in the BIM project created. Please provide evidence on the integration of 3D and analysis data into BIM.
RESULTS
3.1. Digitisation and modelling
From the written content and the described approach, it is understood that the BIM creation was based on ideal/theoretical/abstracted geometries from TLS point cloud data measurements, i.e., without representing the as-built or as-is condition of the building as in some of the publications suggested for the Introduction section. Although failing in the representation of the case study's geometrical alterations, a schematic BIM project can be suitable for data integration and sharing among stakeholders. For the building itself (above ground level), it seems to be the case; therefore, in this manuscript, this ideal 3D modelling reach/scope should be reflected on.
Further, it is not described how the Mudejar coffered-ceiling dome was modelled. Please describe.
Please note that all this and content included in subsection 3.1 is purely methodological. Please, move it to the MATERIALS AND METHODS section and keep for RESULTS just research outcomes, e.g., Figures 9-11.
3.3. Integration...
This subsection describes the methodology in two phases. Please move that to the MATERIALS AND METHODS section.
3.4. Quantitative analysis and visualization
This is just visualisation. Please provide tables with quantitative data on the analyses carried out.
Also, in this sense, one expects that the BIM integration is complete, in such a way that quantitative data are integrated in the HBIM by linking them with (BIM) 3D geometry.
LANGUAGE
British and US English should be unified. Please be consistent; e.g., "digitization" (line 114) and "modelling" (line 127).
Please revise the use of verb tenses. Sometimes, it is written in present simple; others, in past simple using passive voice.
Comments on the Quality of English LanguageBritish and US English should be unified. Please be consistent; e.g., "digitization" (line 114) and "modelling" (line 127).
Please revise the use of verb tenses. Sometimes, it is written in present simple; others, in past simple using passive voice.
Author Response

(The authors gave the same response as above.)

Round 2
Reviewer 1 Report
Comments and Suggestions for Authors
The revised manuscript can be considered for publication.
Comments on the Quality of English LanguageN/A
Author Response
Dear reviewer,
Thank you for your positive feedback and for taking the time to review our revised manuscript. We are delighted that you consider it suitable for publication. Without any doubt your comments and suggestions have significantly contributed to the improvement of our work. We appreciate your efforts and support throughout the review process.
The authors,
Reviewer 2 Report
Comments and Suggestions for Authors
Dear Authors,
Thank you for submitting the improved version of your manuscript. I think it presents a much higher level of scientific soundness than its previous form. I also appreciate your answers to my comments and concerns.
To sum up, I do not see any other objections and recommend your text for publishing.
Congratulations!
Author Response
Dear Reviewer,
Thank you very much for your kind words and for taking the time to review our revised manuscript. We are happy that you find the improved version to be of a higher scientific standard and that our responses have adequately addressed your comments and concerns.
Your feedback has been really useful in refining our work, and we are grateful for your recommendation for publication.
Thank you once again for your support and encouragement.
Best regards,
The authors,
Reviewer 3 Report
Comments and Suggestions for Authors
Authors addressed all my comments and justified those they found not applicable.
Only a couple of minor details drew my attention:
- Point clouds are not converted into 3D meshes. Instead, 3D meshes are modelled, computed, based on point clouds. The reason for this is that mesh vertexes do not generally coincide with points in clouds (do not have exactly the same coordinates); rather, those 3D meshes are an approximation to the point clouds. Triangulation algorithms aim to fit discretised geometries to the point cloud. To "fit" is not to "connect point-cloud points using triangles". Hope that explanation helps.
- The access date of reference [1] should be adapted to Author's case.
- As Authors are writing in US English, please refer to "modeling" instead of "modelling", the latter being British spelling.
Comments on the Quality of English Language- As Authors are writing in US English, please refer to "modeling" instead of "modelling", the latter being British spelling.
Author Response
Dear reviewer,
Thank you for your valuable feedback and for pointing out the minor details that need attention.
- As for the first comment on point clouds and 3D meshes, we appreciate your detailed explanation. To clarify, in our methodology, we used calibrated spheres with known radius, we fit that spheres to create a block adjustment, ensuring precise alignment. The word “convert” in the S281 "This process involved the application of advanced mesh reconstruction algorithms to convert the point cloud data into a structured representation of the Pavilion's surface," was used to avoid redundancy since "modelling" appears both before and after this sentence… Actually it continues “Once the mesh modelling was completed…” We hope to clarify this.
- We have updated the access date for reference [1] to reflect our case.
- We have also corrected the spelling to "modeling" to maintain consistency with US English throughout the manuscript. We have also corrected others similar cases such as “analyse”, “endeavours”, “behaviour”, “honour”…
Thank you again for your insightful comments and for helping us improve our manuscript.
Best regards,
The authors,